# How Does Abusive Supervision Affect Organisational Gossip? Understanding the Mediating Role of the Dark Triad [note 1]

**DOI:** 10.3390/bs13090730

**Published:** 2023-08-31

**Authors:** Fatih Uçan, Salih Börteçine Avci

**Affiliations:** 1Faculty of Economics and Administrative Sciences, Atatürk University, Erzurum 25050, Turkey; savci@atauni.edu.tr; 2Master Araştırma Eğitim ve Danışmanlık Hizmetleri Ltd., Şti., Ata Teknokent, Erzurum 25050, Turkey

**Keywords:** organisational gossip, abusive supervision, dark triad (DT), structural equation modelling (SEM), educational organisations

## Abstract

According to the trait activation theory (TAT), personality characteristics are dormant until contextual elements stir them into action. Personality traits are expected to be activated in the context of abusive supervision. From this perspective, our paper examines whether abusive supervision affects organisational gossiping behaviour through the dark triad. To this end, this study examines the mediating effects of the dark triad on the relationship between abusive supervision and organisational gossip based on cross-sectional data gathered from two separate samples. Using the results from structural equation modelling, it is evident that abusive supervision activates the dark triad, and its context influences organisational gossip in line with the TAT. In addition, our results show that abusive supervision positively affects gossip for information gathering and relationship building, with the dark triad proving to be completely mediating. This finding implies that abusive supervision is a contextual factor, and as such, behaviours such as consistent ill treatment and non-violent, verbal or non-verbal hostile acts will have long-term and lasting effects on organisational communication in many organisations. This study offers significant policy implications concerning behavioural issues within education-centred organisations.

## 1. Introduction

Gossip has received more attention as of late as an inevitable fact and phenomenon within organisational communication networks [1,2,3]. Gossip is often used by employees in the workplace, either as a strategic and political tool or as a means of seeking relief and responding to social injustice [4]. Gossip can affect policies with contradictory roles at the organisational level. In particular, employees gossip during their official duties due to the instrumental ties between them that arise [5] and within private relationships when they are with friends or giving advice to one another [6]. Regarding its negative role, gossip is associated with organisational issues, such as lost reputations, social undermining, decreased productivity, and wasted time [7,8,9].

On the other hand, gossip has positive roles in the organisation, such as information acquisition, relationship building, social enjoyment and group protection, and organisational citizenship behaviours [10,11,12]. Organisations are affected by the positive and negative aspects of gossip, and this effect can play a significant role in determining organisational development and human resource policies. The underlying motives for gossiping behaviour involve many different motives, such as information gathering, relationship building, social enjoyment, and having a negative influence on others [13]. This study focuses on three types of gossip: information gathering, relationship building, and negative influence. Information gathering gossip exchanges are evaluative (positive or negative) information about third parties [12]. Relationship building gossip builds relationships or strengthen relationships between gossipers via the subject of the person who is gossiped about [14]. Negative influence gossip is the influence of others and manipulation of their opinions, typically negatively [15].

Gossip in an organisational setting is determined and sustained by organisational possibilities that regulate and govern the survival of the individual and the group [16]. While most of the current studies in the literature have focused on determining the managerial-level consequences of gossip in organisations [17,18,19], there are few studies on how gossip starts [2]. Most studies that explain the structural antecedents of gossip in organisations and determine the reasons for participating in gossip at the individual level point to the organisational context [7,20]. Thus, the contextual nature of gossip [9] makes it an intriguing phenomenon worth investigating in live situational settings [2]. In particular, an organisational context involving abusive supervision as a stressor [21] provides an opportunity for an in-depth examination of behaviours.

Abusive supervision refers to a perceptual representation of the psychological state induced by a manager in his or her subordinates [22]. Previous research has reported differing views on the effects of abusive supervision on employees’ well-being, attitudes, and behaviour. Some have expressed the adverse effects of abusive supervision on subordinates and the negative reactions of subordinates [23,24,25], while others believe that subordinates are not always negatively affected and will not react negatively [26,27,28]. For this reason, a call for a more balanced perspective through studies examining the differing effects of abusive supervision on subordinates is expressed in the literature [28]. Despite opposing views, studies confirm that subordinates’ reactions to deliberately unexpected behaviour by a supervisor will vary depending on individual and situational characteristics [23,25].

The connection between contextual factors and individual traits is debated through the trait activation theory (TAT). The choice of this theory as the framework for this study stems from its potential to enhance our understanding of how internal psychological mechanisms are triggered when an abusive supervisor creates a hostile environment and puts subordinates in a vulnerable position [29,30]. Tett and Burnett [31] argue that personality traits are only behaviourally active once contextual factors activate them. These factors include daily tasks, social demands from working with supervisors, and organisational culture [32].

Trait activation may occur due to social demands from peers, subordinates, customers, and supervisors [31]. TAT is broad enough to be applied to various personalities, which could fuel more systematic research on person–situation interactionism in predicting behaviours such as organisational gossip [30]. Considering the current debates, personality is seen as a phenomenon that interacts with many other factors that affect human behaviour [33]. Considering that gossip is, to some extent, personality dependent [9,13], personality may explain the psychological mechanism of the relationship between abusive supervision and these types of gossip. Various models for understanding personalities include HEXACO, the Myers–Briggs Type Indicator, the 16 personality factors, and the Big Five personality traits [34]. Moreover, there are three personality traits, Machiavellianism, narcissism, and psychopathy, on the dark side of relating, which is an inevitable part of interacting with others [35]. It is reported that individuals with a dark triad are more ready to gossip for their own good, engage in selfish actions, and ignore norms [13]. Furthermore, dark triads (narcissism, Machiavellianism, and psychopathy) may be activated due to subordinates’ negative emotions. They may seek supportive activities (such as information gathering and relationship building) under the direction of these traits, and gossip can help this process.

Researchers distinguish between intense situations with normative expectations and clear roles that limit behaviour and relaxed situations that allow for more freedom. It has been argued that one’s behaviour in intense situations is based on situational conditions rather than individual personality [13,36]. Therefore, this study included two educational institutions, a public high school and a public university, to explore the relationships between the variables for a deeper understanding. Educators working in public high schools, where quite intense situations are assumed to occur, must adjust their behaviour to comply with the rules and achieve organisational goals. On the other hand, educators at public universities are subject to fewer regulations and guidelines and regulate their conduct based on their own internalized standards [37].

Based on these explanations and to the best of our knowledge, this study investigates the influence of abusive supervision on organisational gossip for the first time and examines the potential mediating role of the dark triad from the perspective of TAT for educational organisations. This study addresses the existing research gap by examining three different types of organisational gossip and making several contributions to the literature. First, this study re-examines the internal mechanism that leads subordinates to gossip when faced with abusive supervision and offers a new perspective for understanding the relationship between the two. Second, this paper employs Machiavellianism, narcissism, and psychopathy as mediator variables. When integrated into the model separately, the dark triad supports a deeper understanding of the internal mechanism by mediating the relationship between abusive supervision and organisational gossip. Thirdly, based on trait activation theory, this study provides an in-depth understanding of how subordinates, who use personality traits triggered by negative situations as a tool, resort to organisational gossip to cope with abusive supervision. Thus, it helps to understand how the dark triad of employees acts as a buffer when facing negative situations. Consequently, this research has contributed significantly to the existing literature by thoroughly examining examples in educational organisations.

This paper is structured as follows: Section 2 provides an overview of the conceptual framework and hypothesis relationships. The modelling, data sources, and methods are explained in Section 3. In Section 4, the empirical outcomes and discussion are indicated. The last section evaluates the empirical results and proposes organisational policies for educational organisations.

## 2. Conceptual Framework and Hypothesized Relationships

### 2.1. The Relationship between Abusive Supervision and Organisational Gossip

Today’s workplace often faces organisational problems created by corporate officials who exploit and abuse employees or employees who engage in deceptive and sneaky business behaviour [23]. Although different labels such as petty tyranny [38], supervisor aggression [39], and supervisor undermining [40] have been used to explain these behaviours of supervisors, most of the studies conducted today use the term ‘abusive supervision’. It is known that abusive supervisors make derogatory comments towards their subordinates, treat them as if they do not exist, do not share important information, and humiliate them in front of others [40,41]. Tepper [22] defines abusive supervision as “subordinates’ subjective perceptions of the continuity of managers’ non-violent, verbal or non-verbal hostile behavior towards their subordinates”. Ashforth [38], on the other hand, describes abusive supervisors as people who use their power and authority callously and arbitrarily to mistreat employees. Empirical research often examines abuse from the subordinate’s point of view [22,24,38,42]. The current study follows this tradition. Many studies reveal that employees exposed to abusive supervision suffer from this situation which imposes negative costs on organisations [43,44]. For example, employees who display withdrawal behaviours resulting from abusive supervision [27], who decide to leave [45], and who have low job satisfaction [46] have been proven to lead to adverse business outcomes, such as poor job performance [47]. Moreover, managerial hostility that does not include physical violence also reduces the quality of one’s work life [39]. 

Although the idea that abusive supervision has many destructive effects is generally prevalent, some studies indicate its possible positive effects. It is claimed that especially when employees encounter abusive supervision, they show constructive reactions, such as problem solving [23]. On the other hand, the study of Zhang and Liu [28] claimed that abusive supervision in the cultural context of the Asia-Pacific region, unlike the West, may be positively related to subordinates’ experiences of anger and promotion-oriented work efforts. In other words, subordinates can protect themselves from the potential effects of abusive supervision by adopting positive rather than negative coping strategies [28]. In line with the Yerkes–Dodson law, studies have shown that individuals can increase their problem-solving and risk-taking behaviour and adapt their decision making when faced with a stressful context [48,49]. While the idea of there being many adverse effects is generally prevalent, a more balanced view of the impact associated with abusive supervision is needed when considering possible positive effects [28]. The best employee behaviour for which the positive and negative effects of abusive supervision can be observed is gossip [17,18,19].

Gossip can be described as sharing information that includes judgments about absent third parties [12]. According to Rosnow [50], gossip is an instrumental process used by individuals to talk with others for various purposes, such as gaining status, amusement, financial gain, acquiring information, or fulfilling specific preconceived desires and anticipations. When we look at why employees gossip, we come across reasons such as facilitating the information-sharing process, creating a social environment, gaining coercive power, creating power dynamics within organisations, and controlling people in the workplace [17]. It is argued that motivations such as verifying information, gathering information, building relationships, and negatively influencing others underlie gossiping [13]. These motives are also validated in terms of their functions at the organisational level [51]. The first two are information gathering [5,16,52,53] and relationship building [11,53,54,55], which are also considered positive gossip types. The third is having a negative influence on others, which we can consider as a negative gossip dimension [9,12,55].

In our literature review, only some studies have examined the relationship between abusive management and organisational gossip. Decoster, Camps, Stouten, Vandevyvere, and Tripp [26], who found that abusive management predicts organisational gossip, emphasize in their study that employees exposed to abusive management tend to gossip about their perceived harmony with their colleagues and their supervisors. Interestingly, employees confronted with an abusive supervisor exhibit a stronger rapport and less gossip when they identify more strongly with their organisation [26]. Naeem, Weng, Ali, and Hameed [17] researched the effect of perceived subordinates, negative workplace gossip, and supervisors’ negative emotions on abusive supervision in China. Their findings corroborated the association between negative workplace gossip and abusive supervision. Ye, He, and Sun [18] examined the impact of subordinates’ negative workplace gossip on abusive supervision in China. Their hierarchical regression analysis results confirmed that subordinates’ perceived negative workplace gossip is positively related to abusive supervision. In two studies, Ahmad, et al. [56] investigated subordinates’ gossiping behaviour, thoughts of revenge, and abusive supervision. Their outcomes showed that subordinates’ gossiping leads to increased abusive supervision. The existing literature primarily comprises studies on abusive supervision, and its reported effects on subordinates’ gossip have been both positive and negative.

On the other hand, Beersma, Van Kleef, and Dijkstra [10] reported that when people are in danger from norm violations and have the opportunity to discuss them with a member of the same group, they use gossip to gather and verify information and to protect the group against norm violations [10]. The absence of studies examining the potential implications of abusive supervision for types of organisational gossip has created a research opportunity. Given the utilitarian purpose of gossip, we suggest that when faced with aversive actions from a supervisor, subordinates may react with either positive or negative types of gossip to deal with abusive supervision. As a result, we expect the following:

**Hypothesis** **1a:**
*Abusive supervision positively relates to information gathering gossip.*


**Hypothesis** **1b:**
*Abusive supervision positively relates to relationship building gossip.*


**Hypothesis** **1c:**
*Abusive supervision negatively relates to negative influence gossip.*


### 2.2. The Role of Subordinates’ Dark Triad Personality Traits Based on Trait Activation Theory

Understanding what triggers personality traits, described as the latent potential found in an individual, is crucial to understanding the role of personality in the workplace [31]. Janowski and Szczepańska-Przekota [34] suggested that the personality issue extends throughout all organisational operations involving human resources, representing a distinct form of individual drive within a company. Personality traits, defined as dominant constructs in psychology, require certain situations to become activated and direct behavioural responses [57]. Trait activation theory (TAT) formalizes the trait–state relationship by recognizing that the behavioural expression of a trait requires the constant stimulation of that trait by relevant situational cues [57]. This study argues that personality traits, such as the dark triad, can best explain how employees exposed to maladministration perceive this situation and the degree of its reflection on their behaviour [13] and suggests the conceptual model in Figure 1. Although it shares a common core with callous manipulation [58] and is considered socially undesirable, the dark triad differs and has distinct defining features [58,59]. Although these traits deserve to be clustered, their correlations are typically relatively modest, so each can be viewed as a particular aspect of socially aversive behaviour [60]. Narcissists believe they have the right to exploit others for their own benefit [61,62]. Machiavellians are characterized by apathy, the strive for Argentine goals (i.e., money, power, and status), calculating, cunning tactics of manipulation, and a cold lack of emotion and ethical concern [63]. Machiavellians maximize their self-interest through deception and disregard for others [64]. At the subclinical level, psychopaths are impulsive, seek excitement, lack empathy, lead an irregular lifestyle, and are likely to exhibit anti-social behaviours [65]. In summary, ego identity goals drive narcissistic behaviour, instrumental goals drive Machiavellian and psychopathic behaviour, and all three have an emotionless core that encourages interpersonal manipulation [66].

Traits constitute behavioural probabilities that can be explained according to situational demands [67,68]. The principle of trait activation theory also holds that personality traits are expressed as responses to trait-relevant situational cues [31]. Trait activation becomes the process of individuals expressing their traits when particularly relevant situational cues are presented. For example, take a situation in which someone shouts for help when they see that someone else needs help. This situation indicates that the person has the trait of compassion. Otherwise, unresponsiveness means low compassion [31]. Considering that abusive supervisors make derogatory comments towards their subordinates, pretend they do not exist, do not share important information, and humiliate them in front of others [40,41], not every subordinate exposed to this situation will give the same reaction. For example, an employee with high honesty (i.e., low on Machiavellianism and psychopathy) will not perceive embezzlement as an option, regardless of need or opportunity [68]. Lyons and Hughes [69] explored the possible effects of narcissism, psychopathy, and Machiavellianism on the multiple functions of gossip. In their study, psychopathy and narcissism were associated with social pleasure, group protection, and negative influence gossip; Machiavellianism was found to be associated with negative influence gossip. The assertive and belligerent nature of an abusive supervisor can hinder employee status/career development by making employees feel stupid and humiliating them [22]. Abusive supervisors can mobilize narcissists’ distrust of others. One study found that narcissists who felt threatened (compared to those who were not) responded by expressing a significantly negative view of the personality of others [61]. Whereas charismatic and superior narcissists believe other people should be interested in their work [70].

Due to their position in the hierarchy, managers can create situations that will affect employees positively or negatively [71]. Therefore, some factors may cause the abuse of employees who encounter abusive supervision, while others may suppress this behaviour. Gossip is significantly linked to an individual’s personality and the context in which this information exchange occurs [9]. The literature remains silent on whether the characteristics of subordinates who encounter abusive supervision are predisposed to similar gossip behaviours. However, Tepper [25] claims in his conceptual model that the characteristics and behaviours of employees have a regulatory effect on the adverse effects of abusive supervision on employees. In other words, Tepper’s conceptual model reflects that abusive supervision does not affect all subordinates similarly. Based on these explanations, the employees’ personality traits may mediate the relationship between abusive supervision and organisational gossip since not every employee encountering abusive supervision will react similarly in the study. In that sense, this study explores the following hypotheses.

**Hypothesis** **2a:**
*Narcissism positively mediates the relationship between abusive supervision and information gathering gossip.*


**Hypothesis** **2b:**
*Narcissism positively mediates the relationship between abusive supervision and relationship building gossip.*


**Hypothesis** **2c:**
*Narcissism negatively mediates the relationship between abusive supervision and negative influence gossip.*


The unethical nature of Machiavellianism is revealed when there are clues about traits [30]. According to the TAT, the expression of Machiavellian behaviour depends on contextual cues from the psychological work environment. Specifically, it is claimed to be activated by situational cues that are activated or align with the exploitative nature of Machiavellians [72]. Therefore, the impulse control of the Machiavellian is triggered when malicious leadership, such as abusive supervision, creates a risky situation [68,70]. Greenbaum, Hill, Mawritz, and Quade [30] found that abusive supervisors strengthen the relationship between Machiavellianism and unethical behaviours by giving clues that activate the Machiavellian employee’s distrust of others, immoral manipulation, and the desire for control and status. As a result, we would expect the following:

**Hypothesis** **3a:**
*Machiavellianism positively mediates the relationship between abusive supervision information gathering gossip.*


**Hypothesis** **3b:**
*Machiavellianism positively mediates the relationship between abusive supervision and gossip relationship building gossip.*


**Hypothesis** **3c:**
*Machiavellianism negatively mediates the relationship between abusive supervision and negative influence gossip.*


Like other dark triad traits, psychopathy is critical in explaining how traits are activated against situations [73]. It is claimed that psychopathy is not taxonomic (i.e., present or absent) but dimensional (i.e., it varies in degree among individuals) [29]. In contrast to narcissists, individuals with high psychopathic tendencies are less sensitive to risk and related losses [70]. Abusive supervisors can create external, powerful threats that psychopaths cannot directly control. In particular, managers who threaten the interests/expectations of their subordinates will activate the psychopathic features of their employees [29]. When dissatisfied with the situation created by the leader, followers with higher psychopathic traits may have the opportunity to break the rules and question the status quo [74]. As a result, we would expect the following:

**Hypothesis** **4a:**
*Psychopathy positively mediates the relationship between abusive supervision and information gathering gossip.*


**Hypothesis** **4b:**
*Psychopathy positively mediates the relationship between abusive supervision and relationship building gossip.*


**Hypothesis** **4c:**
*Psychopathy negatively mediates the relationship between abusive supervision and negative influence gossip.*


## 3. Materials and Methods

### 3.1. Study Design, Participants, and Procedures

This study, which we conducted based on the exploratory research approach, is deductive research in which primary data are used with a quantitative research design. A structured questionnaire covering 68 variables (79 with demographic variables) was used to analyse cross-sectional quantitative data collected simultaneously. Questionnaire can be seen in the Appendix A. Pen-and-paper questionnaires were distributed face-to-face to the individuals during working hours by a member of the research team between February and May 2021. The ethics committee report was shown to all participants in advance, and it was assured that their answers would be evaluated anonymously. The researcher returned to collect the questionnaires two days after the day he distributed them.

Data were collected through stratified random sampling (SRS), a probability sampling method. SRS is a modification of random sampling in which the target population is divided into two or more relevant strata based on one or more attributes [75]. To achieve greater generalizability and to present a constructive replication of the findings [76], the hypotheses were tested with two different samples: (a) teachers working in public high schools and (b) academic staff working at a public university. We focused on educators since a single director generally governs educational organisations, plans with a distinct goal in mind, is limited to a certain period, has a clearly outlined and organised syllabus which is given by specially qualified staff, and follows rigorous disciplinary standards, unlike other organisations. Thus, it provides us with the opportunity to examine a context created by abusive administration more accurately. To avoid having multiple respondents from the same school/faculty, we asked for an equal number of respondents from each to participate in the survey.

Moreover, collecting data from multiple sources at multiple points in the same period reduces concerns that our findings are affected by common method bias. This multigroup SEM aims to determine whether the study is acceptable for the two proposed cultural samples. To clarify whether the type of gossip is related to the personality of the gossiper, a sample from public schools and a second sample from a public university were used. Respondents did not receive any incentives or compensation for their participation in the survey. The research procedure was applied equally for both samples.

*(S1) Sample 1.* S1 is composed of 470 teachers working in high schools operating under the Ministry of National Education affiliated with the central government in Turkey. The sample included 261 women and 209 men with an average age of 34.9 (SD = 0.81) years. Three hundred forty-two of the participants (72.8%) were married. The participants had been working for their current organisation for 5.5 (SD = 1.37) years. In regards to the participants’ workloads according to their statements, eighty-five (18.1%) were very busy, 136 (28.9%) were busy, 222 (47.2%) were moderately busy, 22 (4.7%) were not very busy, and 5 (1.1%) were not at all busy.

*(S2) Sample 2.* The participants in S2 were 990 academic staff from a public university operating in the same region. The respondents were primarily male (*n* = 668, 67.5%) with an average age of 43.2 years (SD = 0.99). Seven hundred and five of the participants (71.2%) were married. The participants had been working in the department for an average of 9.2 (SD = 1.58) years. In regards to the participants’ workloads according to their statements, two hundred seventy-eight (28.1%) were very busy, 325 (32.8%) were busy, 351 (35.5%) were moderately busy, 29 (2.9%) were not very busy, and 7 (0.7%) were not at all busy. As a result, a total of 1460 responses were obtained from two samples for this study.

### 3.2. Measures

Constructs were measured using multi-item scales from existing studies whenever possible. However, some items have been slightly modified to fit the research context. All items were on a 5-point Likert scale from 1 = strongly disagree to 5 = strongly agree, and the scales were used similarly in both samples.

*Abusive Supervision*: A 15-item version of the scale of abusive supervision perception by Tepper [22] was used for the present study. The sample item for the scale is “Supervisor ridicules me”. The reliabilities assessed by Cronbach’s alpha values indicated good and acceptable reliabilities, respectively (Sample 1 α = 0.93 and Sample 2 α = 0.95).

*Dark Triad*: The scale of Jonason and Webster [77] was used to measure the dark triad. The scale consists of three sub-dimensions, namely narcissism (sample item: I tend to want others to pay attention to me), Machiavellianism (sample item: I tend to manipulate others to get my way), and psychopathy (sample item: I tend to lack remorse). The Cronbach’s alpha reliability coefficients for the overall scale were α = 0.86 (Sample 1) and α = 0.84 (Sample 2).

*Organisational Gossip*: A 24-item revised version of the organisational gossip scale developed by Han and Dağlı [51] was used. The scale consists of three sub-dimensions. They are distributed in the following manner: information gathering (sample item: I learn many things about my colleagues at my organisation through gossip), relationship building (sample item: Gossiping with my colleagues at my organisation increases our sincerity), and negative influence (sample item: The gossip in my organisation causes disagreements among us). The reliability of the overall scale was good: α = 0.89 (Sample 1); and α = 0.84 (Sample 2).

### 3.3. Control Variables

Individual differences are reported to predict attitudes towards gossip and the tendency to gossip [78]. In particular, age and gender could predict organisational gossip [9,79,80], and were controlled in the tests conducted with both samples. The correlation matrix of means, standard deviations, and scale reliability for all variables is given in Table 1 and Table 2.

### 3.4. Data Analysis

We tested our hypotheses using a structural equation modelling (SEM) framework with a mediation model. The same analyses were conducted for both samples [81,82]. Following the two-stage approach of Anderson and Gerbing [83], first, confirmatory factor analysis (CFA) was used to confirm the fit of the measurement model to the data. Then regression analysis and path analysis were performed on the structural model. Descriptive analyses were performed to examine the preliminary correlations between the main variables. The study also measured Cronbach’s alpha and McDonald’s omega. The mediating effects of narcissism, Machiavellianism, and psychopathy were tested by the bootstrapping procedure [84] with 5000 sampling to generate 95% confidence intervals. Mediation effect analysis is proposed to determine whether a mediation effect exists and the type of mediation’s corresponding parameters which refer to influence affect value. The mediating effect is considered statistically significant when the bootstrap 95% confidence interval does not include 0. The goodness of fit of the measurement and structural model was evaluated using the following indicators: chi-square/df, goodness-of-fit index (GFI), normed fit index (NFI), incremental fit index (IFI), comparative fit index (CFI), and root mean square error of approximation (RMSEA) [85]. Additionally, we checked for two types of error: incorrect sign (Type S) and incorrect magnitude (Type M) for all hypotheses. Gelman and Carlin [86] provide a simple function to estimate these errors in their paper called retro design. The Type S error rate is “the probability that the replicated estimate has the incorrect sign if it is statistically significantly different from zero”. The exaggeration ratio (expected Type-M error) is “the expectation of the absolute value of the estimate divided by the effect size, if statistically significantly different from zero” [86]. The noisier or more variable the data, the less one should be confident about any inferences based on statistical significance, as illustrated with the Type-S (size) and Type-M (magnitude) errors [86,87,88].

### 3.5. Construct Diagnostics

In both samples, confirmatory factor analysis (CFA) showed an acceptable measurement pattern (χ^2^/df = 2.13 (S1); 3.11 (S2), GFI = 0.86 (S1); 0.85 (S2), NFI = 0.89 (S1); 0.88 (S2):, IFI = 0.91 (S1); 0.91 (S2), CFI = 0.90 (S1); 0.91 (S2), RMSEA = 0.049 (S1); 0.046 (S2). The χ^2^ statistics were significant, and all fit indices were within the recommended acceptable ranges [89]. All standardized first-order loadings were positive and statistically significant, suggesting that the factors were well-defined in both samples.

The measurement model generally has two validity and two reliability measures. Convergent validity and discriminant validity are used for validity. In contrast, composite reliability, Cronbach’s alpha [90], and McDonald’s omega assess internal consistency, for which values equal to or higher than 0.700 were considered good [91]. Convergent validity refers to the degree to which multiple measures of a construct that should theoretically be related are related [92]. Convergent validity is assessed using average variance extracted (AVE), which shows how much of the variance of the indicators can be explained by the latent unobserved variable. All factor loading values are in the range of 0.625–0.878 (S1) and 0.605:0.872 (S2); therefore, factor loadings for both samples exceed the threshold limit of 0.6. AVE values are 0.511–0.665 (S1) and 0.568:0.689 (S2), which are higher than the recommended cut-off value of 0.5 [93]. These results show that the model has good convergent validity for both samples. Discriminant or divergent validity indicates how much a particular construct differs from others. It refers to how different the metrics, which should not be highly correlated, actually are [83].

The discriminant validity is measured by the Fornell and Larcker ratio criterion (Table 3) and the Heterotrait–Monotrait (HTMT) (Table 4) correlation ratio. The Fornell and Larcker ratio criterion measures discriminant validity in which all diagonal values in the table must be greater than the underlying values. HTMT technique examines the ratio of between-trait correlations to within-trait correlations to the correlations of indicators within a construct [94]. Table 2 shows that all values written in bold on the diagonal are more significant than those below. This result makes the Fornell and Larcker ratio criteria results appropriate. Additionally, the second measure to estimate discriminant validity is the HTMT ratio, for which all values must be less than 0.90. The table shows that all values are less than the threshold point. Thus, the discriminant validity condition was also met.

When the reliability results were evaluated, the composite reliability (CR) values calculated according to the factor loads were 0.79–0.95 (S1); they varied between 0.76 and 0.96 (S2). The Cronbach’s alpha values and McDonald’s omega values were greater than 0.7, which is similarly acceptable [91,95]. According to these results, reliability was confirmed.

### 3.6. Common Method Variance

Common method variance, defined as artificial correlation [96] among constructs attributable to the measurement method, can distort survey-based results (bias). We investigated common method bias using Harman’s single-factor test and used an unrotated principal component analysis with varimax rotation [96]. It was seen that each item was loaded on its theoretical structure and hidden common method factor. Additionally, the mean variance extracted by the common method factor was 0.27, well below the 0.50 threshold that [97] associated with an independent construct. Therefore, common method bias is not overly problematic for this study.

## 4. Results

The descriptive analysis results and correlations of abusive supervision, narcissism, Machiavellianism, psychopathy, information gathering gossip, relationship building gossip, and negative influence gossip are presented in Table 1 and Table 2. Abusive supervision was found to be significantly positively related to narcissism, Machiavellianism, psychopathy, information gathering gossip, and relationship building gossip but was significantly negatively related to negative influence gossip. Meanwhile, narcissism was positively and significantly related to Machiavellianism, psychopathy, information gathering gossip, relationship building gossip, and negative influence gossip. Machiavellianism was significantly and positively related to psychopathy, information gathering gossip, and relationship building gossip but negatively related to negative influence gossip. Psychopathy was positively related to information gathering gossip and relationship building gossip but negatively related to negative influence gossip.

The complete structural model shown in Figure 1 was evaluated separately for both samples. A mediation analysis explored how abusive supervision is related to information gathering, relationship building, and negative influence gossip. The bootstrapping method was used to test the existence of mediating effects, as recommended by Hayes [98]. Bootstrapping is used in mediational analyses to generate an empirically derived representation of the sampling distribution of the indirect effect and then bootstrapped confidence intervals. The results of the 5000-bootstrap sample at the 95% confidence interval are shown in Table 5, Table 6 and Table 7. If the 95% confidence limits include zero, the indirect effect test is insignificant at the 95% confidence interval. Table 5, Table 6 and Table 7 present the results of the mediation models used to test the hypotheses. These three tables examine the direct and indirect effects of information gathering, relationship building, and negative influence gossip, which are the types of organisational gossip. As they represent different types of gossip, the hypothesis tests for each type of organisational gossip were analysed separately. Using the obtained effect size estimate and the standard error for all hypothesis relationships, we calculate the Type-S and Type-M errors with the help of Rstudio (alpha = 0.05 is the default significance level, and n.sims = 10,000 is the default number of simulations). As Gelman and Carlin [86] point out, we ensured the Type-S (size) error rate was less than 0.5, and the Type-M (magnitude) error rate was greater than 1 in each of our analyses.

### 4.1. Mediated Regression Results for Information Gathering Gossip

A mediation analysis explored how abusive supervision is related to information gathering gossip. First, the results showed that the model adequately fit the data for both samples (S1: χ^2^/df = 2.16 GFI = 0.880; NFI = 0.940; CFI = 0.950; RMSEA = 0.50)–(S2: χ^2^/df = 3.38; GFI = 0.885; NFI = 0.901; CFI = 0.951 RMSEA = 0.049). Second, this study employed *p*-values, beta values, confidence intervals, and Type-S and Type-M errors to confirm the statistical significance and the direction were positive, and the results for the structural model assessment are illustrated in Table 5. H1a was tested and confirmed as there was a positive and significant association between abusive supervision and information gathering gossip. As shown in Table 5, the direct effect of abusive supervision on information gathering gossip was not statistically significant for sample 1 (β = 0.085, *p* = 0.086, 95% CI [−0.013, 0.187] S/M = 0.144/6.264) and there was a significant effect for sample 2 (β = 0.111, *p*= 0.001, 95% CI [0.041, 0.180]; S/M = 0.042/3.625).

Three indirect effects (mediating variables) were introduced and are proposed in Table 5. All of these three mediating roles were accepted for both samples. Hypotheses H2a, H3a, and H4a confirmed that narcissism, Machiavellianism, and psychopathy positively mediated the relationship between abusive supervision and information gathering gossip. Narcissism significantly and positively mediated the relationship between abusive supervision and information gathering gossip for both samples (S1: β = 0.076, *p* = 0.014, 95% CI [0.014, 0.180] S/M = 0.102/5.192; S2: β = 0.047, *p* = 0.001, 95% CI [0.021, 0.082] S/M = 0.001/1.919). Machiavellianism significantly and positively mediated the relationship between abusive supervision and information gathering gossip for both samples (S1: β = 0.074, *p* = 0.050, 95% CI [0.002, 0.204] S/M = 0.048/3.783; S2: β = 0.036, *p* = 0.001, 95% CI [0.014, 0.071] S/M = 0.006/2.348) Psychopathy significantly and positively mediated the relationship between abusive supervision and information gathering gossip for both samples (S1: β = 0.186, *p* < 0.001, 95% CI [0.070, 0.300] S/M = 0.139/6.077; S2: β = 0.052, *p* < 0.001, 95% CI [0.019, 0.110] S/M = 0.299/12.899). As a result, the higher the level of abusive supervision, the higher the level of narcissism, Machiavellianism, and psychopathy, leading to better information gathering gossip. Therefore, an indirect effect test indicated that narcissism, Machiavellianism, and psychopathy (fully mediated in sample 1 and partially mediated in sample 2) mediated the relationship between abusive supervision and information gathering gossip. The effects of the control variables are shown in Table 5. The direct effect of the gender and age of the subordinates on information gathering gossip was not statistically significant for both samples. Furthermore, the Type-S rates, which indicate that, based on our sample size, the risk that the sign indicates the effect we observe is incorrect, is low, and the Type-M rates, which indicate that, based on our sample size, there is not any risk that we are overestimating the magnitude of the significant effect uncovered for all significant hypotheses.

### 4.2. Mediated Regression Results for Relationship Building Gossip

The model fit values showed an acceptable fit in Table 6, which showed the situation in which relationship building gossip is the dependent variable (S1: χ^2^/df = 2.18; GFI = 0.858; NFI= 0.941; CFI = 0.953 RMSEA = 0.050), (S2: χ^2^/df = 3.41; GFI = 0.887; NFI= 0.912; CFI = 0.967 RMSEA = 0.049). The H1b is tested and confirmed that a positive and significant association between abusive supervision and relationship building gossip exists. The results are presented in Table 6. The direct effect of abusive supervision on relationship building gossip was not statistically significant for both samples (S1:β = 0.055, *p* = 0.255, 95% CI [−0.050, 0.167] S/M = 0.186/7.418; S2:β = 0.039, *p* = 0.241, 95% CI [−0.031, 0.112] S/M = 0.039/3.567).

Hypotheses H2b, H3b, and H4b confirmed that narcissism, Machiavellianism, and psychopathy positively mediated the relationship between abusive supervision and relationship building gossip. Narcissism significantly and positively mediated the relationship between abusive supervision and relationship building gossip for both samples (S1: β = 0.069, *p* = 0.011, 95% CI [0.015, 0.161] S/M = 0.069/4.341; S2: β = 0.027, *p* = 0.001, 95% CI [0.011, 0.051] S/M = 0.003/1.969). Machiavellianism significantly and positively mediated the relationship between abusive supervision and relationship building gossip for both samples (S1: β = 0.101, *p* = 0.012, 95% CI [0.012, 0.249] S/M = 0.034/3.416; S2: β = 0.065, *p* < 0.001, 95% CI [0.036, 0.108] S/M = 0.001/1.068). Psychopathy significantly and positively mediated the relationship between abusive supervision and relationship building gossip for both samples (S1: β = 0.149, *p* = 0.001, 95% CI [0.052, 0.313] S/M = 0.151/6.440; S2: β = 0.088, *p* < 0.001, 95% CI [0.044, 0.154] S/M = 0.010/1.797). The higher the level of abusive supervision, the higher the level of the dark triad, leading to better relationship building gossip. Therefore, an indirect effect test indicated that narcissism, Machiavellianism, and psychopathy fully mediated the relationship between abusive supervision and relationship building gossip for both samples. The effects of the control variables are shown in Table 6. The direct effect of the gender and age of the subordinates on relationship building gossip was not statistically significant for both samples. The Type-S rates are low, indicating there is a low risk that the sign of the effect we observed is incorrect, and the Type-M rates indicate that there is no risk that we are overestimating the magnitude of the significant effect uncovered for all significant hypotheses.

### 4.3. Mediated Regression Model for Negative Influence Gossip

A mediation analysis was conducted to explore how abusive supervision is related to negative influence gossip. First, the results showed that the model adequately fit the data for both samples (S1: χ^2^/df = 2.06; GFI = 0.857; NFI= 0.942; CFI = 0.952 RMSEA = 0.048)–(S2: χ^2^/df = 3.14; GFI = 0.887; NFI= 0.890; CFI = 0.922 RMSEA = 0.047). Our second hypothesis, Hypothesis 1c, predicted that abusive supervision is negatively related to negative influence gossip. An examination of Table 7 reveals that we did not find statistically significant support for Hypothesis 1c; abusive supervision was not positively related to negative influence gossip in either of the samples (S1: β = −0.051, *p* = 0.339, 95% CI [−0.150, 0.055] S/M = 0.474/109.339; S2: β = 0.001, *p*= 0.985, 95% CI [−0.073, 0.072] S/M= 0.373/21.564).

Hypotheses H2c, H3c, and H4c confirmed that narcissism, Machiavellianism, and psychopathy negatively mediated the relationship between abusive supervision and negative influence gossip. Contrary to our expectations, narcissism significantly and positively mediated the relationship between abusive supervision and negative influence gossip for sample 1 (S1: β = 0.053, *p* = 0.011, 95% CI [0.009, 0.146] S/M = 0.004/2.157). However, there was no significant indirect effect for sample 2 (S2: β = 0.009, *p* = 0.230, 95% CI [−0.006, 0.031] S/M = 0.015/2.764). Machiavellianism significantly and negatively mediated the relationship between abusive supervision and negative influence gossip for both samples (S1: β = −0.106, *p* = 0.016, 95% CI [−0.018, −0.261] S/M = 0.479/132.381; S2: β = −0.036, *p* = 0.003, 95% CI [−0.072, −0.011] S/M = 0.281/11.756). The higher the level of abusive supervision, the higher the level of Machiavellianism, leading to more negative influence gossip. Psychopathy significantly and negatively mediated the relationship between abusive supervision and negative influence gossip for sample 2 (β = −0.035, *p* = 0.003, 95% CI [−0.081, −0.011] S/M = 0.328/15.352). However, there was no significant indirect effect for sample 1 (β = −0.054, *p* = 0.263, 95% CI [−0.233, 0.045] S/M = 0.328/15.352). Therefore, an indirect effect test indicated that the dark triad fully mediated the relationship between abusive supervision and negative influence gossip for sample 1. The effects of the control variables are shown in Table 7. The direct effect of the gender of the subordinates on the negative influence of gossip was not statistically significant for either of the samples. The direct effect of age on the negative influence of gossip was statistically significant for sample 2 but not for sample 1. Furthermore, the risk that the sign of the effect we observed is incorrect is low, and there is not any risk that we are overestimating the magnitude of the significant effect uncovered for all significant hypotheses.

## 5. Discussion

Based on trait activation theory, the current study deepened our understanding of the role of trait factors (i.e., the dark triad) in organisational gossip by clarifying how abusive supervision is associated with information gathering, developing relations, and negative influence gossip. While many studies have investigated the effect of gossip on abusive supervision [17,18,28,56], the current study is the first to examine whether abusive supervision affects organisational gossip types and the mediating effect of dark triad characteristics on this relationship. A total of 470 participants from sample 1 and 990 from sample 2 within educational organisations were investigated for our empirical analysis of the proposed hypotheses, and our results demonstrate the following. (1a) Abusive supervision favourably influences information gathering gossip for both studies. (2a, 3a, 4a) Narcissism, Machiavellianism, and psychopathy play a mediating role in the relationship between abusive supervision and information gathering gossip for both studies. (1b) The direct effect of abusive supervision on relationship building gossip is not statistically significant for both studies. (2b, 3b, 4b) Narcissism, Machiavellianism, and psychopathy play a mediating role in the relationship between abusive supervision and relationship building gossip. (3a) The direct effect of abusive supervision on negative influence gossip is not statistically significant for either of the studies. (2c) Narcissism is a mediating factor in the relationship between abusive supervision and negative influence gossip for sample 1. However, it is not significant for sample 2. (3c) Machiavellianism is a mediating factor in the relationship between abusive supervision and negative influence gossip for both studies. (4c) Psychopathy plays a mediating role in the relationship between abusive supervision and negative influence gossip for sample 2 but is not significant for sample 1.

To begin with, the findings of this study indicated a low occurrence of abusive supervision in both samples. One potential explanation for this trend is the study’s location in Turkey, a country with a high power distance score (66). In such an environment, hierarchical superiors, often perceived as paternal figures, tend to be distant and inaccessible [99]. High power distance, one of the distinctive cultural values in Turkey, indicates that employees there might respond differently to malicious supervision compared to those in countries with low power distance. Thus, the present study allowed for a richer assessment of abusive supervision that takes cultural contexts into account [28].

The results indicate that abusive supervision has a significant positive effect on information gathering gossip. This finding contradicts the assumption that “reactions to interpersonal threats are always negative” [100]. Given that gossip often serves as an information gathering tool, subordinates may engage in gossip to gain insight into navigating and mitigating exposure to a supervisor’s abusive behaviour or how to respond to such attitudes [50]. Similarly, studies that posit employees who encounter abusive supervision will gossip to seek information [55,101] align with this finding. However, it is worth noting that the direct effects of abusive supervision on relationship building gossip and negative influence gossip are not statistically significant. These findings contrast with those of previous studies [17,18,26].

Our study contributes to the development of trait activation theory (TAT). TAT posits that specific work-related factors influence an individual’s psychological tendencies and that the influence of personality traits may vary depending on the context [57]. These factors include aspects such as the task itself (e.g., daily tasks, responsibilities, and procedures), social dynamics (e.g., social demands arising from interactions with peers, supervisors, subordinates, and customers) and organisational factors (e.g., organisational culture, climate, and structure) [32]. This theory furthers our knowledge of how abusive supervision affects the types of organisational gossip and the conditions that activate employees’ dark triad features. This study also demonstrates the significance of the dark triad in the relationship between supervisors and employees. The results indicate that abusive supervision influences the tendencies of organisational gossip, mediated by the presence of narcissism, Machiavellianism, and psychopathy. This result is consistent with Chaby, Sheriff, Hirrlinger, and Braithwaite [49]’s findings and the Yerkes–Dodson law. Chaby et al. [49] highlight the importance of context in behaviour and performance, stating that in the case of fear conditioning, behaviour changes according to the Yerkes–Dodson law. Considering that an abusive supervisor creates a threatening environment and risky situation for subordinates [25], there are social conditions (such as abusive supervision) in which the dark triad can be activated. Exposure to workplace mismanagement predisposes subordinates to seek solutions to get them out of the situation and to develop complicated and hostile personalities that reflect a faster coping strategy (such as gossip). The study also examines the factors that activate employees’ personality traits, starting with the conditions created by abusive management, and makes a significant contribution to research on employees’ gossiping behaviour.

Michelson and Mouly [9] claim that few published studies reveal personality-related factors that affect the frequency or nature of gossip. There are different reasons why people may be motivated to engage in information gathering, relationship building, and negative influence gossip [12,17,69,102]. Firstly, narcissists, who often have an extreme love of self, a grandiose sense of self-importance, and a powerful sense of entitlement, are motivated to maintain their identity and legitimacy [103]. Consistent with Pertuz-Peralta et al. [104]’s findings, this study is quite interesting as it shows that narcissistic subordinates activated by abusive supervision, characterized by a sense of greatness and need for admiration, tend to gossip, which will increase information sharing with other subordinates. Characterized by amoral manipulation, a distrust of others, the desire for control, and the desire for status, Machiavellians may perceive they are threatened by others in their pursuit of maximizing their interests. A strong position would be a good option [105]. Our findings suggest that, unlike the findings of Liu [106], Machiavellians turn to gossip to gather information when faced with abusive supervision. Although there is a consensus that psychopathy is an underlying factor in deviant interpersonal behaviours that cause distress for co-workers [107], results show that psychopathy explains the relationship between abusive supervision and information gathering gossip. People high in psychopathy generally resist stress, including interpersonal abuse, and seem to need fewer positive relationships than others [108].

Our findings suggest that subordinates tend to be more impulsive, callous, and manipulative when exposed to a supervisor’s behaviour. Therefore, our findings are consistent with those of Jones and Paulhus [109], who stated that narcissism, Machiavellianism, and psychopathy traits are associated with exploitative interpersonal behaviour. Insensitive to the concerns and social restrictions of others, narcissists carry out most of their identity-targeting efforts in the social sphere [61]. Our mediation analysis supports those with the narcissistic personality trait facing abusive supervision will strive to maintain a positive self-image at all costs. The findings of Smalley and Stake [110] show that they suppress hostility by showing positive gossip tendencies (relationship building), as opposed to the finding that narcissism is associated with hostility versus humiliation and hostility. Consequently, our study demonstrates that narcissists can utilize the impact of abusive control to legitimize organisational gossip mechanisms.

Inconsistent previous studies [30,32] showed that Machiavellians are more willing to behave socially when they experience abusive supervisor behaviour. Machiavellians, who are more manipulative towards others, may view stressful situations as potentially harmful, believing they can exploit them if they cannot manipulate others [111]. Our study shows that abusive supervision can easily provoke Machiavellianism. This finding may be associated with Machiavellians being more willing to provide support and learn. As a result, when faced with a difficult situation, Machiavellians are willing to gather information to manipulate others or provide social support to meet their interests. The finding that psychopathy mediates the relationship between abusive supervision and developing relations gossip confirms that people with psychopathic tendencies can better cope with stressful situations [112]. Moreover, findings suggest that individuals with high levels of psychopathy remain unaffected by abusive behaviours.

We should note the unexpected results produced for narcissism and psychopathy’s mediating role. Interestingly, our results showed a significant difference in the mediated role of narcissism and psychopathy between abusive supervision and negative influence gossip for both studies. Narcissism was an intermediary between abusive supervision and negative influence gossip for sample 1, but not for sample 2. Psychopathy was an intermediary between abusive supervision and negative influence for sample 2, not for sample 1. The differences in our findings across sample 1 and sample 2 may be due to differences in organisational cultures and differences in the operationalization of gossip behaviours. This finding supported that the logic that aversive behaviour, narcissism, Machiavellianism, and psychopathy are related to bad things. The finding that Machiavellianism has a negative mediating effect for negative influence gossip supports the explanation that Machiavellians exhibit corporate citizenship behaviours that can lead to positive social impressions [113]. Another possible reason is that as they identify with their organisation; employees with a stronger sense of harmony with their workgroup buffer their tendency to gossip about their leaders negatively to protect the organisation’s image [26]. That is, even though employees who value their organisation may be inclined to participate in gossip for negative influence when confronted with an abusive supervisor, they will be less inclined to gossip to protect the organisation’s image from potentially harmful consequences. These findings support studies claiming that employees who identify with their organisations experience more favourable outcomes regarding well-being and behaviour, even in challenging and persistent personal contexts [114,115].

Furthermore, it was found that gender was not associated with information gathering, developing relations, and negative influence gossip. The analysis yielded consistent results for both samples. The high school teachers in sample 1 were not accessible during their school hours. However, they were free before or after their teaching hours, whereas the academics in sample 2 were more independent than the teachers [116]. The nature of these occupations meant that these two samples were not biased in their gender composition. These unique sample characteristics may explain why, for example, gender was not a significant control variable in this study. Finally, the results indicated that age did not associate with developing relationships. However, our results indicated a significant difference in the association between age and information gathering and the negative influence of gossip for both samples. The probable reason for this result is the difference in the average age of the samples (Sample 1 had an average age of 34.9 and Sample 2 had an average of 43.2). Those in Sample 2 were almost a decade older with lots more experience than those in Sample 1. Despite the limited literature, our findings differ from Massar, Buunk, and Rempt [80]’s study. Massar et al. [80] found a significant negative effect between a participant’s age and their tendency to gossip. The difference in experience between the two samples may be a possible reason for this result.

## 6. Limitations and Future Direction

This study provides suggestions for future research while recognizing its inherent limitations. The first limitation stems from the cross-sectional nature of this study, which prevents us from drawing conclusions on the nature of causality [117]. Further research using experimental or longitudinal designs may help clarify these issues. Second, the sample of this study was limited to those working in the education sector, which may raise concerns about the generalizability of our findings. While this is a concern, it is also important to note that two samples from the education sector provided a reasonably suitable testing ground for our hypotheses, suggesting that the effects found were powerful. With further research, these findings must be replicated in groups of subjects from various industries. Examining the effect of positive personality traits to support TAT in future gossip studies is recommended. Exploring gossip from different perspectives has the potential to provide practitioners with a better understanding of this critical phenomenon. The findings of this study can be evaluated more broadly in studies conducted in different cultures.

## 7. Conclusions and Policy Implications

The dark side of management and unregulated organisational communication systems endangers the sustainability of organisations and the peace climate within organisations. This research examines the types of organisational gossip from the perspective of trait activation theory in educational organisations by applying structural equation modelling. In this context, the current research sheds light on the literature on organisational behaviour by integrating the effects of abusive supervision on organisational gossip, considering the mediating role of the dark triad. The results show that abusive supervision affects information gathering gossip but not relationship building gossip or negatively influence gossip. Moreover, the empirical evidence further confirms the mediating role of narcissism, Machiavellianism, and psychopathy on the relationship between abusive supervision and the three categories of organisational gossip. Furthermore, the results of this study indicate that abusive supervision may serve as a situational factor that activates narcissism, Machiavellianism, and psychopathy and predicts information.

Based on these findings, this study offers policy suggestions for educational organisations in particular and for all organisations in general. Even if educators have some autonomy in their school/faculty, their share of participatory activities in management should be improved. Last, the findings demonstrated that despite the ramifications of abusive supervision, a sense of trust, organisational identification, and perceptions of organisational justice can significantly affect organisational gossip. Consequently, human resource policies that enhance employees’ personal characteristics and organisational identity should be adopted, management–employee relationships strengthened, and perceptions of organisational justice increased.

## Figures and Tables

**Figure 1 behavsci-13-00730-f001:**
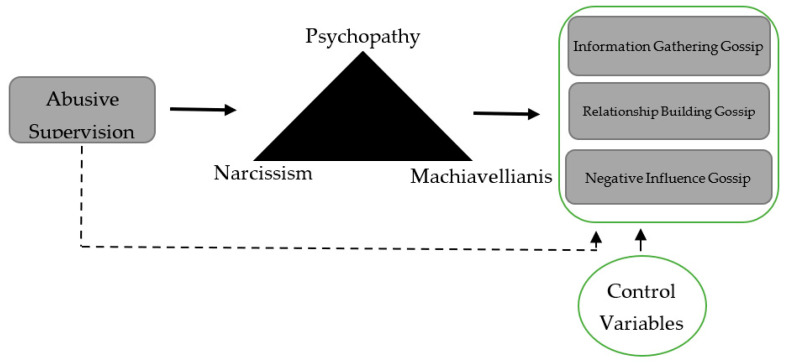
The conceptual model.

**Table 1 behavsci-13-00730-t001:** Reliabilities and factor loadings.

		Sample 1 (*n* = 470)					Sample 2 (*n* = 990)					
Construct	Mean	SD	CR	AVE	ω	CFA Range	Mean	SD	CR	AVE	ω	CFA Range
Abusive Supervision	1.16	0.38	0.94	0.51	0.93	0.63–0.81	1.47	0.72	0.96	0.59	0.95	0.70–0.84
Narcissism	2.61	0.99	0.90	0.55	0.88	0.65–0.78	2.92	0.90	0.87	0.60	0.86	0.61–0.75
Machiavellianism	1.38	0.59	0.89	0.62	0.86	0.63–0.88	1.49	0.71	0.89	0.62	0.87	0.67–0.86
Psychopathy	1.70	0.71	0.79	0.53	0.77	0.66–0.76	1.93	0.72	0.76	0.59	0.71	0.68–0.74
İnformation Gathering	2.14	1.05	0.93	0.66	0.94	0.72–0.87	2.62	1.09	0.94	0.69	0.94	0.76–0.87
Relationship Building	2.00	0.96	0.92	0.63	0.94	0.74–0.84	1.98	0.91	0.91	0.59	0.91	0.67–0.82
Negative Influence	3.52	1.13	0.95	0.67	0.95	0.73–0.88	3.72	0.93	0.93	0.57	0.92	0.65–0.84
χ^2^/df = 2.13 CFI =0.90 RMSEA = 0.05							χ^2^/df = 3.11 CFI = 0.91 RMSEA = 0.05					

**Table 2 behavsci-13-00730-t002:** Correlations and scale reliabilities.

	1	2	3	4	5	6	7	8	9
Sample 1 (*n* = 470)									
1. Gender	______								
2. Age	−0.32 **	––––––							
3. Abusive Supervision	−0.06	−0.03	**0.93**						
4. Narcissism	0.01	−0.13 **	0.14 **	**0.87**					
5. Machiavellianism	−0.02	−0.07	0.29 **	0.19 **	**0.87**				
6. Psychopathy	−0.06	−0.04	0.28 **	0.36 **	0.52 **	**0.75**			
7. İnformation Gathering	0.06	−0.11 *	0.28 **	0.38 **	0.36 **	0.44 **	**0.94**		
8. Relationship Building	0.07	−0.10 *	0.24 **	0.38 **	0.43 **	0.42 **	0.67 **	**0.94**	
9. Negative Influence	−0.02	0.03	−0.15 **	0.11 *	−0.20 **	−0.13 **	−0.10 *	−0.24 **	**0.94**
Sample 2 (*n* = 990)									
1. Gender	––––								
2. Age	−0.30 **	––––––							
3. Abusive Supervision	−0.04	0.01	**0.95**						
4. Narcissism	0.14 **	−0.13 **	0.15 **	**0.84**					
5. Machiavellianism	−0.02	−0.09 **	0.23 **	0.28 **	**0.87**				
6. Psychopathy	−0.04	−0.08 **	0.22 **	0.32 **	0.45 **	**0.67**			
7. İnformation Gathering	0.08 **	−0.09 **	0.24 **	0.29 **	0.26 **	0.31 **	**0.94**		
8. Relationship Building	0.02	−0.06	0.21 **	0.29 **	0.40 **	0.44 **	0.51 **	**0.91**	
9. Negative Influence	−0.05	0.14 **	−0.03	−0.05	−0.20 **	−0.16 **	−0.13 **	−0.37 **	**0.92**

** *p* < 0.05; * *p* < 0.10; Cronbach’s alpha (in the diagonal).

**Table 3 behavsci-13-00730-t003:** HTMT ratio.

Sample 1							Sample 2					
	1	2	3	4	5	6	1	2	3	4	5	6
1. Machiavellianism	-						-					
2. Narcissism	0.18	-					0.22	-				
3. Psychopathy	0.63	0.36	-				0.43	0.27	-			
4. Abusive Supervision	0.27	0.10	0.22	-			0.20	0.12	0.15	-		
5. Information Gathering	0.32	0.41	0.43	0.21	-		0.24	0.28	0.35	0.19	-	
6. Relationship Building	0.36	0.42	0.47	0.20	0.69	-	0.38	0.26	0.31	0.17	0.48	-
7. Negative Influence	−0.14	.15	−0.12	−0.08	−0.02	−0.19	−0.15	−0.03	−0.26	−0.04	−0.09	−0.30

**Table 4 behavsci-13-00730-t004:** Fornell and Larcker criterion.

Sample 1								Sample 2						
	1	2	3	4	5	6	7	1	2	3	4	5	6	7
1. Machiavellianism	**0.79**							**0.77**						
2. Narcissism	0.22	**0.74**						0.28	**0.77**					
3. Psychopathy	0.70	0.46	**0.73**					0.62	0.41	**0.79**				
4. Abusive Supervision	0.28	0.11	0.25	**0.72**				0.23	0.15	0.21	**0.77**			
5. Information Gathering	0.35	0.42	0.52	0.22	**0.81**			0.29	0.33	0.36	0.21	**0.83**		
6. Relationship Building	0.39	0.42	0.52	0.21	0.70	**0.79**		0.45	0.32	0.56	0.19	0.54	**0.77**	
7. Negative Influence	−0.16	0.11	−0.13	−0.10	−0.04	−0.22	**0.82**	−0.18	−0.04	−0.20	−0.05	−0.11	−0.35	**0.75**

Correlations among constructs compared with the square root of the AVE (in the diagonal).

**Table 5 behavsci-13-00730-t005:** Mediated regression results of information gathering gossip.

	Sample 1 (*n* = 470)						Sample 2 (*n* = 990)					
Pathways	Std.β	SE	*p*	95% CI		Type S/M	Std.β	SE	*p*	95% CI		Type S/M
				Lower	Upper					Lower	Upper	
**Direct Effect**												
Abusive Supervision → Information Gathering (H1a)	0.09	0.11	0.09	−0.01	0.19	0.14/6.26	0.11	0.05	0.00	0.04	0.18	0.04/3.63
Abusive Supervision → Narcissism	0.12	0.11	0.02	0.02	0.22		0.15	0.04	<0.001	0.07	0.24	
Narcissism → Information Gathering	0.30	0.05	<0.001	0.19	0.41		0.24	0.04	<0.001	0.16	0.32	
Abusive Supervision → Machiavellianism	0.29	0.07	<0.001	0.15	0.44		0.24	0.03	<0.001	0.15	0.33	
Machiavellianism → Information Gathering	0.12	0.08	0.02	−0.02	0.25		0.12	0.05	<0.001	0.04	0.20	
Abusive Supervision → Psychopathy	0.27	0.08	<0.001	0.12	0.42		0.22	0.03	<0.001	0.12	0.32	
Psychopathy → Information Gathering	0.32	0.09	<0.001	0.16	0.47		0.19	0.06	<0.001	0.08	0.30	
Gender → Information Gathering	0.06	0.09	0.20	−0.04	0.14		0.06	0.07	0.07	−0.01	0.12	
Age → Information Gathering	−0.03	0.05	0.45	−0.15	0.08		−0.07	0.03	0.02	−0.15	0.02	
**Specified Indirect Effect.**												
Abusive Supervision → Narcissism → Information Gathering (H2a)	0.08	0.04	0.01	0.01	0.18	0.10/5.19	0.05	0.02	0.00	0.02	0.08	0.01/1.92
Abusive Supervision → Machiavellianism → Information Gathering (H3a)	0.07	0.05	0.05	0.00	0.20	0.05/3.78	0.04	0.01	0.00	0.01	0.07	0.01/2.35
Abusive Supervision → Psychopathy → Information Gathering (H4a)	0.19	0.08	0.00	0.07	0.30	0.14/6.08	0.05	0.02	0.00	0.02	0.11	0.30/12.90
**Total Effect**	0.24	0.05	<0.001	0.13	0.34		0.22	0.03	0.00	0.15	0.28	
	χ2/df = 2.16 CFI = 0.95 RMSEA = 0.05						χ^2^/df = 3.38 CFI = 0.95 RMSEA = 0.05					

**Table 6 behavsci-13-00730-t006:** Mediated regression results of relationship building gossip.

	Sample 1 (*n* = 470)						Sample 2 (*n* = 990)					
Pathways	Std.β	SE	*p*	95% CI		Type S/M	Std.β	SE	*p*	95% CI		Type S/M
				Lower	Upper					Lower	Upper	
**Direct Effect**												
Abusive Supervision → Relationship Building (H1b)	0.06	0.10	0.26	−0.05	0.17	0.19/7.42	0.04	0.04	0.24	−0.03	0.11	0.04/3.57
Abusive Supervision → Narcissism	0.12	0.11	0.02	0.02	0.22		0.15	0.04	<0.001	0.07	0.24	
Narcissism → Relationship Building	0.30	0.04	<0.001	0.20	0.41		0.17	0.03	<0.001	0.09	0.25	
Abusive Supervision → Machiavellianism	0.29	0.07	<0.001	0.15	0.44		0.24	0.03	<0.001	0.15	0.33	
Machiavellianism → Relationship Building	0.18	0.07	<0.001	0.02	0.34		0.27	0.04	<0.001	0.17	0.36	
Abusive Supervision → Psychopathy	0.27	0.08	<0.001	0.12	0.43		0.22	0.03	<0.001	0.12	0.33	
Psychopathy → Relationship Building	0.26	0.08	<0.001	0.13	0.45		0.38	0.06	<0.001	0.26	0.50	
Gender → Relationship Building	0.05	0.08	0.24	−0.04	0.14		−0.01	0.06	0.63	−0.07	0.05	
Age → Relationship Building	0.02	0.05	0.67	−0.10	0.14		0.04	0.03	0.18	−0.04	0.12	
**Specified Indirect Effect.**												
Abusive Supervision → Narcissism → Relationship Building (H2b)	0.07	0.04	0.01	0.02	0.16	0.07/4.34	0.03	0.01	0.00	0.01	0.05	0.00/1.97
Abusive Supervision → Machiavellianism → Relationship Building (H3b)	0.10	0.06	0.01	0.01	0.25	0.03/3.42	0.07	0.02	0.00	0.04	0.11	0.00/1.07
Abusive Supervision → Psychopathy → Relationship Building (H4b)	0.15	0.07	0.00	0.05	0.31	0.15/6.44	0.09	0.03	0.00	0.04	0.15	0.01/1.80
**Total Effect**	0.22	0.06	0.00	0.10	0.33		0.21	0.04	0.00	0.13	0.30	
	χ2/df = 2.18 CFI = 0.95 RMSEA = 0.05						χ^2^/df = 3.41 CFI = 0.97 RMSEA = 0.05					

**Table 7 behavsci-13-00730-t007:** Mediated regression results of negative influence gossip.

	Sample 1 (*n* = 470)						Sample 2 (*n* = 990)					
Pathways	Std.β	SE	*p*	95% CI		Type S/M	Std.β	SE	*p*	95% CI		Type S/M
				Lower	Upper					Lower	Upper	
**Direct Effect**												
Abusive Supervision → Negative Influence (H1c)	−0.05	0.14	0.34	−0.15	0.06	0.47/109.34	0.00	0.04	0.99	−0.07	0.07	0.37/21.56
Abusive Supervision → Narcissism	0.27	0.11	0.02	0.02	0.21		0.15	0.04	<0.001	0.07	0.27	
Narcissism → Negative Influence	0.17	0.06	<0.001	0.06	0.29		0.05	0.04	0.19	−0.04	0.14	
Abusive Supervision → Machiavellianism	0.29	0.07	<0.001	0.15	0.46		0.24	0.03	<0.001	0.15	0.33	
Machiavellianism → Negative Influence	−0.14	0.11	0.01	−0.27	−0.01		−0.12	0.05	<0.001	−0.21	−0.03	
Abusive Supervision → Psychopathy	0.27	0.09	<0.001	0.11	0.42		0.21	0.03	<0.001	0.11	0.31	
Psychopathy → Negative Influence	−0.08	0.11	0.19	−0.24	0.08		−0.14	0.06	<0.001	−0.24	−0.04	
Gender → Negative Influence	0.02	0.11	0.61	−0.08	0.12		−0.03	0.07	0.35	−0.11	0.04	
Age → Negative Influence	0.03	0.07	0.49	−0.10	0.16		0.12	0.03	<0.001	0.04	0.21	
**Specified Indirect Effect.**												
Abusive Supervision →Narcissism → Negative Influence (H2c)	0.05	0.03	0.01	0.01	0.15	0.00/2.16	0.01	0.01	0.23	−0.01	0.03	0.02/2.76
Abusive Supervision → Machiavellianism → Negative Influence (H3c)	−0.11	0.06	0.02	−0.02	−0.26	0.48/132.38	−0.04	0.02	0.00	−0.07	−0.01	0.28/11.76
Abusive Supervision →Psychopathy → Negative Influence (H4c)	−0.05	0.07	0.26	−0.23	0.05	0.06/4.06	−0.04	0.02	0.00	−0.08	−0.01	0.33/15.35
**Total Effect**	0.27	0.08	0.00	0.19	0.42		−0.05	0.04	0.21	−0.13	0.03	
	χ^2^/df = 2.06 CFI = 0.95 RMSEA = 0.05						χ^2^/df = 3.14 CFI = 0.92 RMSEA = 0.05					

## Data Availability

The raw data supporting the conclusions of this article will be made available by the authors, without undue reservation.

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
