# Peer review of "How Does Abusive Supervision Affect Organisational Gossip? Understanding the Mediating Role of the Dark Triad†"

_behavsci, 2023, doi:10.3390/bs13090730_

Round 1
Reviewer 1 Report
The conclusions are very brief, despite the fact that the essential information is there. Maybe more details in this part would be good.
Author Response
Dear Editor and Reviewers,
We kindly thank you for your inspirational comments for our manuscript to improve its quality.
Based on your suggestions, we tried to make the changes and corrections. Your comments and our responds to the reviewers are kindly shown in the table below. Prior to address your comments we kindly inform you that all the revisions and added parts are highlighted in yellow in the revised version of our manuscript.
Yours truly,
Please see the attachment

Reviewer 2 Report
Have you tried this research in other sectors?
Author Response

(The authors gave the same response as above.)

Reviewer 3 Report
Dear Authors,
Thank you for submitting the good piece of work. Well done!!
Author Response

(The authors gave the same response as above.)

Reviewer 4 Report
The article looks very decent, reasoned and interesting for the reader.
The authors used a structured questionnaire in this study. So it would be better to show this questionnaire to readers in attachements or Study design, participants, and procedures part of the article. This will help to better understand the respondents' answers and, if necessary, replicate a similar study in another country.
Author Response

(The authors gave the same response as above.)

Reviewer 5 Report
It might be published after a minor revision.
Before forwarding their text to the editor, Authors may decide:
- Make minimal improvements (abstract, conclusions); the text will be ready for publishing but may cause some critics.
- Invest some effort and improve the text; the text will be ready for publishing.
- Invest some more effort in reconstructing indicated issues. The text will be a high-quality publication.
Please refer to the appended file, where:
A list of strengths is provided.
The list of defects is formulated.
The use of statistical tools is highly doubtful.
Suggestions for improvements are given.

Author Response

(The authors gave the same response as above.)

Reviewer 6 Report
Dear Authors.
Thank You for a possibility to review this very well written paper, which I have analyzed with a great interest.
Several comments/suggestions of a general nature.
First, I cannot find Your gossip definition in Your study. I can only guess the negative connotation of this term.
In this matter, I’d like to see the questionnaires. I do not know, what the questions were and what behaviors/messages were perceived as gossips. How the educators identify the source and authors of gossip? You are “walking on thin ice”….
I do not think it was the good idea to implement the terms such as “dark personality”, “bad manger”, “harmful behavior” (what constitutes these terms?). In the context of performance not every “deliberately harmful behavior by a bad manager” (lines 52-53) are always “bad “ for the organization, as a whole system.
I strongly recommend to read:
- https://www.mdpi.com/1996-1073/15/13/4533
- https://doi.org/10.1080/19420889.2015.1029689
and cite in the study. Otherwise, Your conclusions can be perceived as biased from the ground.
Moreover, why did not implement OCEAN model in Your research (of Costa and McCrae). This approach is more general and, as such, more suitable for this kind of study (there are not “negative” and “positive” traits – compare as above). If there are “dark traits”, “light traits” should also exist in an individual dually….
Line 66-67. Do You have any proof of such a statement or this is the postulate only (where is the source?) ?
There are too many repetitions in Your study, even twice in one sentence (f.e. lines 59-60, and many more)
Line 153 – 154. What is the novelty of Your paper?
Line 162 (Yerkes-Dodson rule)
Line 180-181 (f.e. motivation?)
Line 204 - the gossip is innocent (maybe guilty, as charged ) Maybe it is a good idea to correct the content by a native speaker?
Lines 316-317 – this is why I recommend to use Big5 model.
Conclusions
Where is the value added? It seems like "old wine in a new bottle"....
Many repetitions, common language. The consultation with a native speaker highly recommended.
Author Response

(The authors gave the same response as above.)

Round 2
Reviewer 6 Report
Better now.
The English in the paper is better now.